# Valorization of the Salmon Frame as a High-Calcium Ingredient in the Formulation of Nuggets: Evaluation of the Nutritional and Sensory Properties

**DOI:** 10.3390/foods13111701

**Published:** 2024-05-29

**Authors:** Camila Matamala, Paula Garcia, Pedro Valencia, Alvaro Perez, Manuel Ruz, Leyla Sanhueza, Sergio Almonacid, Cristian Ramirez, Marlene Pinto, Paula Jiménez

**Affiliations:** 1Department of Food Science and Chemical Technology, Faculty of Chemical and Pharmaceutical Sciences, Universidad de Chile, Santiago 8380000, Chile; camila.matamala@ug.uchile.cl; 2Department of Nutrition, Faculty of Medicine, Universidad de Chile, Santiago 8380453, Chile; pgarcia@uchile.cl (P.G.); afperezb@uchile.cl (A.P.); mruz@uchile.cl (M.R.); leyla.sanhueza@ug.uchile.cl (L.S.); 3Department of Chemical and Environmental Engineering, Universidad Técnica Federico Santa Maria, Valparaíso 2390123, Chile; pedro.valencia@usm.cl (P.V.); sergio.almonacid@usm.cl (S.A.); cristian.ramirez@usm.cl (C.R.); marlene.pinto@usm.cl (M.P.)

**Keywords:** calcium, salmon frame, bone flour, protein hydrolysate, food ingredient, nuggets

## Abstract

In the Chilean population, calcium consumption is deficient. Therefore, several strategies have been implemented to increase calcium intake, such as consuming dairy products and supplements. In this study, an ingredient composed of bone flour (BF) and protein hydrolysate (PH) obtained from salmon frame was used as an innovative source of calcium. The objective was to evaluate the effect of the incorporation of BF and PH in a 1:1 ratio (providing two calcium concentrations to the nuggets, 75 and 125 mg/100 g) on calcium content and sensory attributes of salmon nuggets submitted to baking or shallow frying. Proximal chemical analyses, fatty acid composition, calcium content, and sensory evaluation (acceptability and check-all-that-apply test) were tested in the nuggets. The incorporation of BF/PH (1:1) in both concentrations increased the calcium content of salmon nuggets being higher for the 125 mg/100 g. On the other hand, no negative effects were observed on sensory properties where all samples showed good overall acceptability for baked and fried nuggets. Therefore, the incorporation of BF/PH (1:1) into salmon nuggets enhances the nutritional quality of these products by providing a higher calcium content without significantly affecting their sensory properties.

## 1. Introduction

The global aquaculture production of Atlantic salmon (*Salmo salar*) in 2020 was approximately 2.7 million tons [1]. Nutritionally, salmon is considered a good source of proteins (17–19%) and long-chain polyunsaturated fatty acids (LC-PUFA), such as EPA (eicosapentaenoic acid, C20:5) and DHA (docosahexaenoic acid, C22:6) [2,3]. In Chile, in 2021, approximately 725,000 tons were produced, which represents 25.1% of the total fish production [4]. It is mainly exported as fresh, chilled, and frozen fillets. The salmon industry has become an important economic activity in Chile, being the main exported product in the year 2021 [5]. Nevertheless, salmon processing generates many byproducts that are mainly disposed of as wastes causing a considerable environmental, economic, and social impact [6]. The main destination of the byproducts of the fishing industry is the production of fishmeal and fish oil. It is estimated that between 27 and 48% of the total production of fishmeal and fish oil are obtained from byproducts [1]. Food byproducts are defined as edible or non-edible parts of a raw material, which within a production process are discarded and do not form part of the final product [7]. According to the type of processing and fish species, it is estimated that between 30 to 70% of the total weight of fish comprises byproducts such as heads (9–12% of total byproducts), viscera (12–18%), skin (1–3%), bones (9–15%), and scales (5%) [1]. At the national level, the percentage of by-products generated by the salmon industry ranges between 15 and 25% of the total fish [7]. However, food byproducts contain various nutrients and bioactive compounds of interest, which could be recovered and utilized in the potential development of functional foods. Regarding salmon byproducts, the frame corresponds to the bones and muscles attached to them [8]. The frame has high protein and mineral contents, mainly calcium and phosphorus, which are found in bones as hydroxyapatite crystals [9,10]. 

Calcium is an important component of bone, and its low consumption is one of the main factors that affect the development of bone mass in children and adolescents and leads to osteoporosis in adults [11]. In the Chilean population, calcium consumption is deficient, being less than 500 mg/day [12] compared to FDA-recommended doses of 1300 mg daily [13]. Although dairy products constitute the main sources of calcium (110 to 1000 mg of calcium/100 g of product), their consumption is low. Similar behavior is observed in the consumption of vegetable foods that provide calcium, such is the case of spinach, broccoli, brussels sprouts, and cauliflower, which contain between 10 to 160 mg of calcium/100 g, as well as nuts and almonds with approximately 75 mg of calcium/100 g [14,15,16]. Therefore, to improve calcium intake in the population, the use of supplements and the incorporation of this mineral have been used in foods. This last strategy has focused on the reformulation of foods with the incorporation of calcium flour from different food byproducts, such as fish frame, fish bones, eggshells, and broccoli flour in foods such as snacks, whole wheat crackers [17], coating of nuggets [18], sausages [19], muffins [20], and tortilla chips [21], among others. In all studies, a significant increase in the calcium content was observed in the food, independent of the origin of the food byproducts.

In this context, a byproduct obtained from salmon bones is its flour. Several studies have reported that it contains 20,200 to 20,400 and 12,500 to 15,100 mg/100 g of calcium and phosphorus, respectively [22], so the frame of salmon could be considered an excellent source of calcium and phosphorus of animal origin. To obtain bone flour (BF) from salmon frame, various treatments have been used: heat treatment with water, enzymatic treatment, and/or chemical treatment. In general, these treatments allow for obtaining high contents of calcium (13,500 to 20,400 mg of Ca/100 g of BF). Higher calcium contents in BF of frames have been obtained through enzymatic hydrolysis (EH) with values between 20,200 and 26,100 mg/100 g [22,23,24]. On the other hand, with the application of the EH process on BF, it is also possible to obtain a protein hydrolysate (PH), which is characterized by a high content of protein (48% of the total crude protein in the salmon frame) and bioactive peptides, such as collagen [8]. Several studies have reported that PH peptides could chelate calcium, allowing for it to be transported to the small intestine, preventing its precipitation in the form of phosphate salts, and therefore increasing its solubility and absorption [25,26].

To our knowledge, in the literature, there are no studies that have described the effects of the incorporation of an ingredient to increase calcium content composed of bone flour (BF) and protein hydrolysate (PH) obtained from the salmon frame. This study was therefore designed to gain new insight into this topic. The aim was to evaluate the effect of the incorporation of BF and PH in a 1:1 ratio (providing two calcium concentrations to the nuggets, 75 and 125 mg/100 g), obtained from salmon frame, on the calcium content and sensory attributes of salmon nuggets submitted to baking or shallow frying.

## 2. Materials and Methods

### 2.1. Materials

Atlantic salmon frame (*Salmo salar*) was donated by Fiordo Austral Company (Puerto Montt, Chile). Salmon fillets were purchased at the local market. Bones flour (BF) and protein hydrolysate (PH) obtained by enzymatic hydrolysis from salmon frame were carried out in the Chemical and Environmental Engineering Department of the Universidad Técnica Federico Santa María (Valparaíso, Chile) according to Valencia et al., 2021 [27]. The sunflower oil, Natura ^®^ (AGD, Córdoba, Argentina), was purchased on the local market. Vegetable fiber CEAMFIBRE 7000K, functional soy protein concentrates WILCON SA, and breadcrumb coating PL10952 CB1 were supplied by DIMERCO Company (Santiago, Chile). Granular sodium tripolyphosphate, antioxidant CMD, and batter 1050 were supplied by CRAMER Company (Santiago, Chile).

### 2.2. Methods

#### 2.2.1. Elaboration of Salmon Nuggets

The elaboration of salmon nuggets was performed in the Laboratory of Nutritional Biochemistry of the Department of Nutrition of the Universidad de Chile. The food ingredient was prepared by mixing BF and PH in a 1:1 ratio (Figure 1). For the formulation of the nuggets, Atlantic salmon fillets, ice, salt, vegetable fiber, concentrated soy protein, sodium tripolyphosphate, and antioxidants were used. For the breading of the nuggets, a mixture of cornstarch, guar gum, and bread was used. Three nugget formulations are shown in Table 1:N: nugget without addition of food ingredient (control);N1: nugget with 0.75% of food ingredient (BF/PH; 1:1);N2: nugget with 1.25% of food ingredient (BF/PH; 1:1).

The incorporation of 0.75 and 1.25% of food ingredient into the nuggets contributed to 75 and 125 mg/100 g of calcium, respectively. The quantity of 75 mg/100 g corresponds to the doses of incorporation of calcium allowed for foods such as fish under Chilean legislation (<10% DRI considering a DRI of 800 mg ca/day) [28], whereas 125 mg/100 g corresponds to FDA recommendations (<10% DRI of 1300 mg of ca/day) [13].

For the elaboration of the nuggets, the salmon fillet and ice were cut in a multiprocessor for approximately 30 s, then all the ingredients were mixed for about 5 min until complete homogenization. In the formulation nuggets with the addition of the food ingredient, salmon fillet (g) was replaced by 0.75 and 1.25% of BF/PH (1:1). The nuggets were molded in a cylindrical shape with dimensions of 5 cm in diameter and 1 cm high and with a weight of 25 g each. Then, they were subsequently frozen at −18 °C. The batter was then prepared, according to the manufacturer’s instructions, and each frozen nugget was dipped in the batter and passed through the breading until the nugget was completely coated. Subsequently, each nugget was subjected to deep pre-frying in sunflower oil in an electric fryer at 170 °C for 1 min; then, the nuggets were cooled to room temperature and, finally, they were frozen and stored at −25 °C until their final preparation (baked or shallow fried).

Nugget formulations:(A)Uncooked or raw nuggets: N (control); N1 (0.75% BF/PH; 1:1); N2: (1.25% BF/PH; 1:1).(B)Fried nuggets: FN (control); FN1 (0.75% BF/PH; 1:1); FN2 (1.25% BF/PH; 1:1). Nuggets were shallow fried using 10 g of sunflower oil Natura ^®^ (AGD, Córdoba, Argentina) in a pan frying at 170–180 °C for 3 min on each side of the nugget.(C)Baked nuggets: BN (control); BN1 (0.75% BF/PH; 1:1); BN2 (1.25% BF/PH; 1:1). Nuggets were baked in an industrial oven at 180 °C for 5 min per side reaching a final internal temperature greater than 80 °C (IPX 5, Electrolux, Porcia, Italy).

#### 2.2.2. Proximal Chemical Analysis

The proximal chemical composition of Atlantic salmon frame, food ingredient (BF/PH, 1:1), raw salmon nugget (N, N1, and N2), baked nugget (BN, BN1, and BN2), and fried nugget (FN, FN1, and FN2) were analyzed. The proximate composition including moisture, proteins, and ash was determined according to the AOAC official procedures [29]. The lipid content was determined using the Bligh–Dyer extraction method (1959) [30], and carbohydrates by difference [31].

#### 2.2.3. Fatty Acids Composition

The fatty acids composition of uncooked salmon nuggets (N, N1, and N2), baked nuggets (BN, BN1, and BN2), and fried nuggets (NF, NF1, and NF2) was performed by gas chromatography. Previously, lipids samples were derivatized to fatty acid methyl esters (FAME) with 20% boron trifluoride (BF_3_) in methanol (*v*/*v*). Then, FAME was analyzed by gas-liquid chromatography (GC) in a GC Agilent (7890A) machine, using a capillary column (Agilent HP—88.60 m × 0.25 mm; i.d.0.25 mm) and a flame ionization detector. C23:0 fatty acid (Nu-Check Prep, Elysian, MN, USA) was used as an internal standard [32].

#### 2.2.4. Determination of Calcium Content

The determination of calcium content of food ingredient (BF/PH, 1:1), uncooked salmon nuggets (N, N1, and N2), baked nuggets (BN, BN1, and BN2), and fried nuggets (FN, FN1, and FN2) was performed by atomic absorption spectrophotometry (Perkin Elmer Analyst 100 atomic absorption spectrophotometer) according AOAC method 985.35 [33]. About 2 g of previously dried nuggets were calcined in a muffle furnace at 500 °C for 8 h. The ashes were then digested with 2 mL of concentrated nitric acid, allowed to evaporate on a hot plate, and placed in the muffle furnace at 500 °C for 1 h. The ashes were then dissolved in 10 mL of 1 N HCl, heated on a hot plate, and transferred to a 10 mL volumetric flask. Additional dilutions were performed with deionized water to bring the concentrations into the linear range of atomic absorption spectroscopy. Lanthanum nitrate was added to the final dilution to achieve a concentration of 0.1% lanthanum.

#### 2.2.5. Sensory Evaluation of Salmon Nuggets

The sensory evaluation was carried out in the Department of Nutrition at the Faculty of Medicine of the University of Chile. This study was approved by the Ethics Committee for the Research in Human Beings (CEISH) of the Faculty of Medicine of the University of Chile on 30 June 2019 (project No. 312-2019). Before the sensory evaluation, the participants signed an informed consent form. Sixty untrained consumers carried evaluations of baked salmon nuggets (BN, BN1, and BN2) and fried nuggets (FN, FN1, and FN2).

Consumer profile: Participants had to indicate their age range (18 to 20; 21 to 25; 26 to 30; 31 to 35; 36 to 40; over 40 years) and their gender (female, male, or other).

Exclusion criteria: The participants had to indicate the food(s) that they did not consume or rejected (beef, potato chips, salmon, dairy products, crackers) and if they had any food allergies. Those who had an allergy to fish and/or did not consume salmon did not participate in this study.

##### Sensory Tests

Acceptability test: The samples were assessed by a panel using the hedonic scale method, with scores ranging from 1 to 7 (7 = extremely like; 6 = very much like; 5 = like; 4 = neither like nor dislike; 3 = dislike; 2 = dislike very much; 1 = dislike extremely), where they had to mark their perception of each attribute (appearance, aroma, flavor, texture, and overall liking). Panel evaluated the baked nuggets (BN, BN1, and BN2) or fried nuggets (FN, FN1, and FN2) separately.

Check-all-that-apply (CATA) questions: In this analysis, a list of terms is provided to consumers, such as bitter taste, fatty flavor, juice, dry texture, weak salmon aroma, strong salmon aroma, light golden color, dark golden color, salty, low salt, crunchy, weak salmon flavor, strong salmon flavor, rough texture, and soft texture. The panel was instructed to select all the words they considered appropriate as descriptors for the samples of both baked (BN, BN1, and BN2) and fried (FN, FN1, and FN2) salmon nuggets.

#### 2.2.6. Statistical Analysis

To assess statistical differences between the results of nutritional composition and sensory evaluation (acceptability test), a one-way analysis of variance (ANOVA) was conducted with a 95% confidence interval (*p* < 0.05) using Statgraphics Centurion XVIII software, and the Tukey HSD multiple range test was applied.

The statistical analysis of the CATA test was carried out using the Cochran Q test with a 95% confidence interval (*p* < 0.05), utilizing XLSTAT version 2021.4 software in Microsoft Excel for Microsoft 365 MSO version 2306.

## 3. Results and Discussion

### 3.1. Proximate Chemical Composition of the Salmon Frame and Food Ingredient (BF/PH)

The proximate chemical composition of the salmon frame and the content of proteins, lipids, and calcium of ingredient (BF/PH; 1:1) are shown in Table 2. The salmon frame presents a high content of proteins, lipids, and ashes (40.0, 46.3, and 14.5 g/100 g, respectively), which are consistent with those reported by other authors with ranges of 33 to 52.3, 35.7 to 59, and 9 to 18 g/100 g, respectively [8,23,34,35]. The differences in the proximate chemical composition of the salmon frame could be attributed to its origin (raw material) and the type of processing applied by the salmon industry. In this context, the EH of salmon frame allowed for obtaining two fractions, BF and PH, constituting an ingredient (BF/PH) with a content of proteins, ashes, and calcium of 35.6 g/100 g, 54 g/100 g, and 10,080 mg/100 g, respectively. Liaset et al., 2003 [8] and Malde et al., 2009 [24] reported similar contents, while other authors have reported lower values of protein (12.07 to 26.6 g/100 g) for salmon bone [22,24]. Lower contents of ash, 43 g/100 g [24], similar contents of 51.4–55 g/100 g [8,24], or higher of 67.8 to 99.9 g/100 g have been reported [22,24]. Regarding calcium content, some authors have described values higher than those found in this study, with a range between 15,700 and 38,800 mg/100 g [8,22,24]. The differences in the content of ash, protein, and calcium in the salmon bone can be explained by the type of hydrolysis used (alkaline or enzymatic) but also should be considered that in the present study, the levels of ashes, proteins, and calcium were determined in the ingredient (BF/PH; 1:1) and not only in salmon bone as in other studies.

### 3.2. Proximate Chemical Composition of Raw, Fried, and Baked Salmon Nuggets

The proximate chemical compositions of raw, fried, and baked salmon nuggets, without (control) and with the incorporation of BF/PH (1:1), are shown in Table 3. The moisture content of salmon nuggets presented a significant reduction (*p* < 0.05) after cooking processes (shallow frying or baking) because these involve the evaporation of the available water in the food [36] (Fellows, 2017) and, therefore, there is an inverse relationship between moisture content and proteins, ashes, carbohydrates, and lipids [36,37]. Fried nuggets (FN, FN1, and FN2) showed significantly lower moisture content compared to baked nuggets (BN, BN1, and BN2). On the other hand, the incorporation of BF/PH (1:1) to the nuggets in the two proportions of calcium (75 mg/100 g and 125 mg/100 g) did not affect moisture content, independent of the cooking process.

Concerning lipid content in raw nuggets, a range of 28 to 31 mg/100 g was observed, which was significantly lower than those found in the fried and baked nuggets, due to the decrease in moisture content resulting from the thermal process they underwent. A significantly higher content (*p* < 0.05) of lipid content was found in fried nuggets (FN, FN1, and FN2) compared to baked nuggets (BN, BN1, and BN2). This can be attributed to the absorption of oil used in the frying process, as the water evaporates, allowing for the oil to penetrate the food [38,39]. This process also occurs when the food is removed from the cooking medium and begins to cool [36,40]. Additionally, when nuggets have coatings, they tend to adsorb oil on their surface [40]. Furthermore, some studies have shown that the increase in lipid content could be associated with the decrease in ash and protein content as occurred in the fried nuggets, compared to the raw and baked ones [41,42].

Concerning protein content, values between 28 to 31 mg/100 g were observed for the raw nuggets, similar to the baked nuggets, and significantly higher in comparison to those found in the fried nuggets. In this regard, it was expected that fried and baked nuggets would have a higher protein content (in comparison to raw nuggets) because of the application of a thermal process that would reduce moisture content and due to the addition of BF/PH containing 36 g/100 g of protein. However, in our study, an increase in protein content in the nuggets was not observed. In the first case, it could be explained by the denaturation of proteins caused by the cooking process, especially frying, resulting in a decrease in the available protein quantity [36] and, secondarily, by the low amount of ingredient added to the nuggets (0.75 and 1.25%). In the sense, Idowu et al. (2019) [26] observed a slight increase in protein content of 11.9 to 12.2 g/100 g, when they substituted 17% of wheat flour with a mixture of biocalcium and protein hydrolysate powders (3:1) from *Salmo salar* frame in whole-wheat cracker.

Regarding ash content, the raw nuggets showed values of 3.6 to 4.3 mg/100 g. This last value was similar in BN2. Also, in all types of nuggets (raw, fried, and baked), it was observed that as the percentage of BF/PH (1:1) addition increased, the ash content in them was higher. This increase is primarily related to the incorporation of the food ingredient with 55 g/100 g of ash. When the nuggets were subjected to shallow frying, the ash content was significantly reduced compared to that of the raw and baked nuggets. These results are opposite to those found by other authors, who have reported that minerals are relatively well preserved when foods are fried at high temperatures within the range of 165–185 °C and for short cooking durations [43,44].

The carbohydrate contents in nuggets primarily originating from the batter (cornstarch) and breading (wheat flour) ranged between 36 and 38 g/100 g for raw nuggets, which were similar to those found in baked nuggets, but significantly higher compared to fried nuggets (35–36 g/100 g). In this latter case, the decrease in carbohydrates could be due to the partial degradation of starchy carbohydrates and the formation of complexes with lipids [36], but also because the frying process increases the formation of resistant starch [44]. On the other hand, the amount of incorporation BF/PH (1:1) in all types of nuggets (raw, baked, and fried) does not show a significant effect on carbohydrate content.

According to the results of the proximal chemical analyses of the nuggets, it can be concluded that, in general, the incorporation of BF/PH (1:1) did not significantly affect their macronutrient content. In contrast, the cooking method, such as shallow frying, did increase the lipid content compared to the baked nuggets.

### 3.3. Determination of Calcium Content of Nuggets

The calcium content in control nuggets (without the addition of BF/PH, 1:1) showed no significant differences between them with values of 14.2, 16.4, and 14.8 mg/100 g for raw, fried, and baked nuggets, respectively. On the contrary, with the incorporation of BF/PH (1:1) (0.75 or 1.25%), an increase in calcium content in the nuggets was observed, which was proportional to the amount of ingredient incorporated. These results are consistent with those reported by other authors, who added flour or powder of bone obtained from fish byproducts to various foods such as pasta [45], bread [46], cookies [47], and crackers [48], increasing their calcium content. Nevertheless, with the incorporation of calcium of 75 and 125 mg/100 g into the nuggets, it was expected to find a final calcium content (raw, fried, or baked nugget) of approximately 90 and 140 mg/100 g, respectively (considering an initial content of calcium in the raw control nugget of 14.2 mg/100 g). Lower contents of calcium of 74 to 86 and 117 to 127 mg/100 g were observed for the incorporation of 75 and 125 mg/100 g, respectively. These lower calcium contents in all nuggets could be attributed to possible losses of the ingredient during the production process. On the other hand, the baked nuggets showed a slightly higher calcium content compared to fried nuggets, considering both levels of ingredient incorporation (0.75 and 1.25%). Segovia, (2014) [49] reported that during the cooking process, calcium may be lost due to its solubility in water that is removed. However, there were no data in the literature that showed a clear trend in the change in calcium content related to the cooking method [50]. According to Reddy and Love (1999) [51], the calcium content is stable at high temperatures, and therefore cooking techniques such as shallow frying and baking should not affect the content of this mineral.

According to these results, it can be established that the incorporation of salmon bones allowed for the calcium content of the nuggets to increase. Thus, the revaluation of byproducts from the food industry and their incorporation into foods could be considered a potential strategy to increase calcium intake in the Chilean population, which is below the established requirements even when there are supplements and frequent recommendations for a healthy lifestyle. A concrete example of the incorporation of micronutrients in foods in our country is given by the recent modification in legislation, which establishes that dairy products and flour must be supplemented with vitamin D [52] due to a vitamin D deficiency in the Chilean population.

### 3.4. Fatty Acid Composition

The fatty acid composition of the lipid fraction of nuggets without and with the ingredient: raw (N, N1, and N2), fried (FN, FN1, and FN2), and baked (BN, BN1, and BN2) are presented in Table 4.

In all types of nuggets (raw or subjected to shallow frying or baking) with or without BF/PH (1:1), it was observed that the fraction of monounsaturated fatty acids (MFAs) was the majority (45.6 to 48%), followed by the fraction of polyunsaturated fatty acids (PFAs) (38.3 to 40.8%), and in a lesser proportion by saturated fatty acids (SFAs) (12.2 to 13.2%). Also, a proportion of 5.5 to 6.4% of trans fatty acids was found. Furthermore, oleic acid (C18:1 n9), linoleic acid (C18:2 n6), and palmitic acid (C16:0) were the main fatty acids identified for MFA (39.5 to 41.4%), PUFA (26.5% to 28.2%), and SFA (8.2 to 9.1%), respectively. On the other hand, in all types of nuggets, the presence of PFA of marine origin, such as eicosapentaenoic acid (EPA, C20:5 n3) and docosahexaenoic acid (DHA, C22:6n3), both characteristic in products made with salmon fillets, ranged from 0.58 to 0.78% for EPA and 1.71 to 2.37% for DHA. The distribution of the fatty acid profile in the nuggets is directly related to the fatty acid composition present in the salmon fillets (the main ingredient in the formulation of the nuggets, at 85%). In our study, it was observed that the main fatty acids found in salmon fillets were oleic acid followed by linoleic acid and palmitic acid with percentages of 24.7%, 12%, and 8.1%, respectively. Additionally, eicosapentaenoic acid (EPA) and docosahexaenoic acids showed percentages of 0.9 and 2.70%, respectively. These values are consistent with those reported by other authors on farmed salmon [53,54] and are different from those found in wild salmon. In the latter, some authors have found percentages of fatty acids in ranges of 13.4 to 17.4% for palmitic acid; 13.3 to 17.1% for oleic acid; 12.9 to 14.6% for DHA; 6.1 to 6.6% for EPA; and less than 1% for linoleic and linolenic acid [54,55]. These differences in the fatty acid composition in farmed salmon could be attributed to the replacement of marine fish oil in salmon feeds with vegetable oils of terrestrial origin, due to the scarcity and increasing price of marine oils. This has led to a change in the biochemical composition of the fish muscle and the specific biochemical composition of farmed seafood, which could be very different from wild salmon due to the formulation of the feeds [54]. Therefore, the high contents of oleic and linoleic acids in all types of salmon nuggets in our study could be attributed to the fatty acid composition of the salmon fillets and the amount of oil absorption during pre-frying of nuggets [36,39,40] carried out with sunflower oil (Natura^®^), which mainly contains linoleic and oleic acid (52.0 and 37.7%, respectively).

Regarding the cooking method, baked salmon nuggets did not show significant differences in the percentages of fatty acids compared to raw nuggets. These results could be attributed to the baking process involving heat transfer through radiation and convection, using temperatures between 110 and 300 °C. However, the heating is slower due to the greater distance between molecules, resulting in lower oxidation [36,56]. In contrast, nuggets submitted to shallow frying with sunflower oil showed a significant decrease (*p* < 0.005) in the total content of saturated (∑SFA) and monounsaturated fatty acids (∑MUFA), while the total content of polyunsaturated (∑PUFA) and trans fatty acid content (∑TransFA) increased (*p* < 0.05), compared to raw and baked nuggets. Results similar to total SFA and PUFA content have been found by other authors, who studied the effect of shallow frying with sunflower oil on the fatty acid composition of fish fillets of seabass [57] and deep frying of tilapia [58]. However, these same authors did not report significant changes in total MUFA content [57,58]. In this context, although it was expected that fried nuggets showed a lower total PUFA content with respect to raw nuggets, due to the high temperature of the process and susceptibility to oxidation of PUFAs [36,56], an increase in this fraction was observed, which could be explained by the significant absorption of linoleic acid in nuggets provided by the sunflower oil used for frying nuggets [53]. In contrast, other PUFAs such as linolenic acid, EPA, and DHA decreased significantly (*p* < 0.05) compared to raw and baked nuggets. Similar results were obtained by other researchers when frying Atlantic salmon fillets [59], seabass [57], and tilapia [58] in sunflower oil. On the other hand, the increase in the fraction of trans fatty acids in the fried nuggets was for the significant increase (*p* < 0.05) in C18:1n9t (elaidic acid). These results contrast with those reported by other authors on fried fish fillets with sunflower oil [57,59]. On the contrary, Yanar et al. (2007) [57] reported a higher content of C18:1n9t in baked fish fillets. Additionally, the incorporation of BF/PH (1:1) with 75 or 125 mg Ca/100 g to salmon nuggets (raw, fried, and baked) did not affect the percentage and composition of fatty acids compared to the control nuggets.

### 3.5. Sensory Evaluation of Salmon Nuggets

#### 3.5.1. Consumer Profile

A total of 107 consumers participated in the sensory evaluation. They evaluated salmon nuggets subjected to baking (*n* = 56) and shallow frying (*n* = 51) without the addition of the food ingredient (control) and with the incorporation of 0.75 and 1.25% of the food ingredient (Figure 2). The baked nuggets were evaluated by 56 consumers (66% women, 21% men, and 13% did not respond); of these participants, 41% were between 21 and 30 years old, and 32% were over 36 years old. The fried nuggets were evaluated by 51 consumers (59% women, 25% men, 4% other, and 12% did not respond) of which 41% were between 21 and 30 years old and 46% were over 36 years old. All participants consumed salmon or fish, and none had any food allergies.

#### 3.5.2. Acceptability Test

The acceptability test was carried out on the samples of baked and fried nuggets separately, where the nuggets without incorporation of BF/PH (control), and nuggets with 0.75% and 1.25% of BF/PH were evaluated for the attributes of appearance, aroma, flavor, texture, and overall acceptability. For baked nuggets, the scores ranged between 5.5 (like) and 6.1 (very much like), whereas fried nuggets ranged between 5.8 (like) and 6.1 (very much like). For baked nuggets, there were no significant differences between the samples (*p* < 0.05) in the attributes of appearance, aroma, and texture. Flavor and overall acceptability showed significant differences but only between the control sample (BN) and the sample with 1.25% of BF/PH (BN2). As was expected, the nugget formulations with higher ingredient content (1.25%) showed lower overall acceptability. Nevertheless, this sample obtained a score of 5.4 and 5.5 for flavor and overall acceptability, respectively, indicating a certain liking (5 = like slightly) rather than dislike. These results coincide with those reported in the literature where some authors indicate that general acceptability tends to diminish when incorporating higher percentages (15 to 20%) of ingredients made from fish byproducts to products such as pasta [60] and bread [46]. For fried nuggets, no significant differences were observed between the samples for any of the attributes studied. According to these results, it is possible to establish that both types of nuggets had good overall acceptability.

#### 3.5.3. Check-All-That-Apply (CATA) Questions

To analyze the results of the CATA test, a correspondence analysis was carried out, which allowed for obtaining a two-dimensional graph that represented 100% of the data variance in the evaluation of baked or fried nuggets (Figure 3 and Figure 4). These graphs allowed for us to analyze the existing relationship between attribute/sample based on the frequency with which each attribute was granted to each sample. Additionally, Cochran’s Q test was performed to determine the existence or absence of significant differences for each attribute between the samples (Table 5 and Table 6).

As can be seen in Figure 3, the baked control nugget was characterized by a strong salmon flavor, strong salmon aroma, slightly salty taste, light gold color, and crispy texture. Baked nuggets with 0.75% BF/PH (BN1) were described as having a rough inside, salty, and dry, and those with 1.25% BF/PH (BN2) were mainly characterized by a soft inside, bitter taste, mild salmon aroma, and juiciness. Although each sample was characterized by different attributes, according to the analysis of Cochran’s Q test (Table 5), it was observed that the frequency of each attribute did not present significant differences between the samples. According to these results, it could be established that the incorporation of BF/PH did not affect the sensory attributes of the baked nuggets, and that the attributes most considered to describe the baked nugget formulations were light gold in color, soft inside, and crispy in texture.

Regarding the fried salmon nugget, in Figure 4, it can be seen that consumers characterized the control sample (FN) by a dark golden color, salty flavor, and rough inside; the nugget with 0.75% BF/PH (FN1) for a mild salmon flavor, crispy texture, juiciness, soft inside, and light gold color; and those with 1.25% BF/PH (FN2) were evaluated with a mild salmon aroma, fatty and slightly salty flavor, and light gold color. However, according to the Q analysis (Table 6), it was observed that only in the juicy and dry attributes were significant differences found between FN, FN1, and FN2. The juicy attribute was more representative of the FN2 sample, while the dry attribute was representative of the FN sample. The remaining attributes did not show significant differences between the samples. Furthermore, Table 6 shows that the attributes most considered in all samples to describe the fried nugget were crispy texture, light golden color, weak salmon aroma, and soft interior.

According to the CATA analysis results, it can be established that for the incorporation levels of the food ingredient used in this study (0.75 and 1.25%), no major differences were observed between the attributes for baked or fried nuggets. The above coincides with the results of the acceptability test, where, in general, all samples presented good overall acceptability.

## 4. Conclusions

Nowadays, the revalorization of food industrial byproducts is of great importance due to their environmental, economic, and social impact. In this research, salmon frame was used as raw material to develop an ingredient with high calcium content composed of BF and PH (1:1). As was expected, the incorporation of this ingredient in baked and fried salmon nuggets improved the nutritional quality of these products by providing higher calcium content without significantly affecting their sensory properties. Considering that the population of our country is deficient in calcium, the development of foods that incorporate new sources of calcium obtained from waste from the food industry could be considered a strategy to increase the intake of this mineral. Furthermore, new studies in humans are necessary that focus on evaluating the absorption and bioavailability of calcium from the consumption of these products.

## Figures and Tables

**Figure 1 foods-13-01701-f001:**
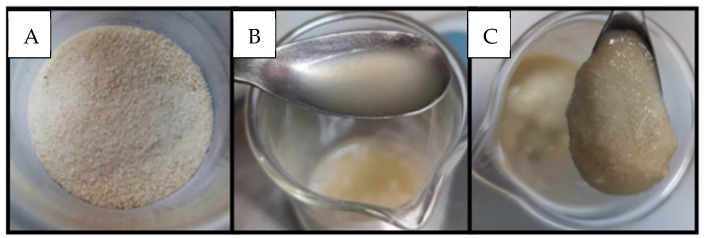
Food ingredient obtained from the salmon frame. BF (**A**), PH (**B**), and BF/PH (1:1) mix (**C**).

**Figure 2 foods-13-01701-f002:**
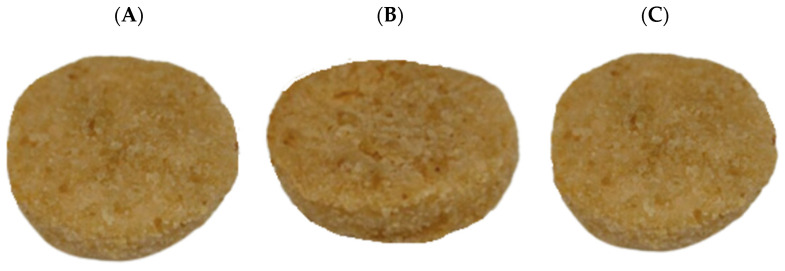
Photography of raw, fried, and baked nuggets. (**A**) BN: baked nugget control; (**B**) BN1: baked nugget + 0.75% BF/PH (1:1); (**C**) BN2: baked nugget + 1.25% BF/PH (1:1); (**D**) FN: fried nugget control; (**E**) FN1: fried nugget + 0.75% BF/PH (1:1); (**F**) FN2: fried nugget + 1.25% BF/PH (1:1); (**G**) raw nugget; (**H**) breaded raw nugget.

**Figure 3 foods-13-01701-f003:**
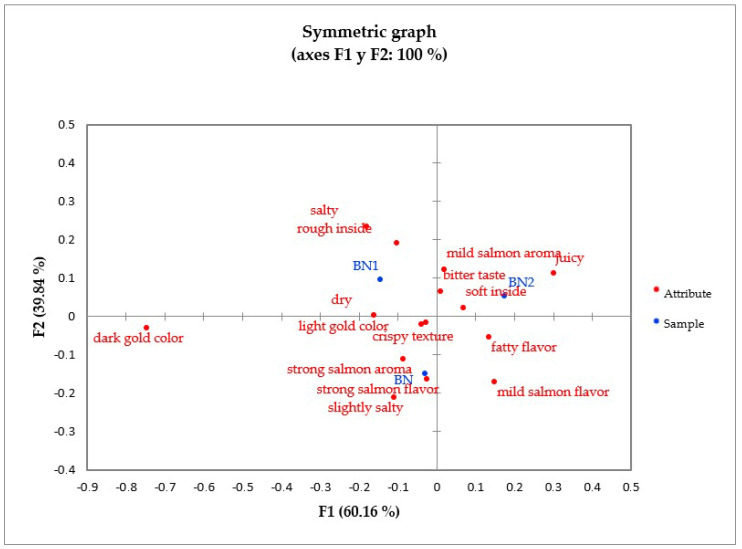
Representation of baked salmon nuggets and terms of the CATA test in the first and second dimensions of the correspondence analysis (*p* < 0.05).

**Figure 4 foods-13-01701-f004:**
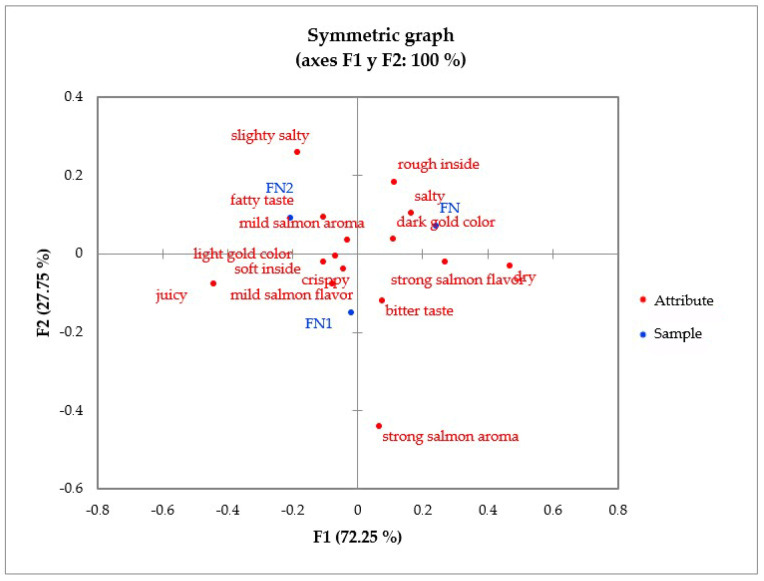
Representation of fried salmon nuggets and terms of the CATA test in the first and second dimensions of the correspondence analysis (*p* < 0.05).

**Table 1 foods-13-01701-t001:** Ingredients of formulations of salmon nuggets (without and with the ingredient BF/HP).

	N	N1	N2
Ingredients (%)			
Salmon fillets	85.6	84.8	84.3
Ice	9	9	9
Vegetal fiber	3	3	3
Concentrated soy protein	1	1	1
Salt	1	1	1
Sodium tripolyphosphate	0.3	0.3	0.3
Antioxidants	0.1	0.1	0.1
BF/PH (1:1)	-	0.75	1.25

BF/PH (1:1): ingredient obtained from the salmon frame by enzymatic hydrolysis and composed of bone flour (BF) and protein hydrolysate (PH).

**Table 2 foods-13-01701-t002:** Proximate chemical composition of the salmon frame and the content of proteins, lipids, and calcium of ingredient BF/PH.

	Salmon Frame (g/100)	BF/PH (1:1) (g/100)
Moisture	54.2 ± 0.8	47.5 ± 0.8
Protein	40.0 ± 0.6	35.6 ± 0.9
Lipid	46.3 ± 1.1	-
Ash	14.5 ± 0.2	54.6 ± 1.3
Calcium (mg/100 g)	-	10,080

BF/PH (1:1): ingredient obtained from salmon frame by enzymatic hydrolysis and composed of bone flour (BF) and protein hydrolysate (PH). The results of moisture, proteins, lipids, and ashes are expressed as X ± SD, g/100 g on a dry basis.

**Table 3 foods-13-01701-t003:** Proximal chemical composition and calcium content of raw, fried, and baked salmon nuggets, without (control) and with the incorporation of BF/PH.

	Uncooked Nugget			Fried Nugget			Baked Nugget		
	N	N1	N2	FN	FN1	FN2	BN	BN1	BN2
Moisture	50.6 ± 0.8 ^d^	50.9 ± 0.8 ^de^	52.7 ± 0.8 ^e^	41.7 ± 0.3 ^a^	42.6 ± 0.8 ^a^	42.3 ± 0.4 ^a^	45.3 ± 0.9 ^b^	47.5 ± 0.7 ^b^	48.3 ± 0.3 ^c^
Lipid	30.9 ± 1.1 ^ab^	28.3 ± 0.6 ^a^	28.0 ± 1.7 ^a^	37.3 ± 1.2 ^d^	36.1 ± 0.5 ^d^	35.9 ± 0.2 ^d^	32.9 ± 1.0 ^c^	32.3 ± 0.4 ^bc^	31.4 ± 0.9 ^bc^
Protein	30.1 ± 0.4 ^d^	29.6 ± 1.0 ^cd^	28.9 ± 1.2 ^d^	25.8 ± 0.1 ^a^	26.2 ± 0.4 ^a^	26.4 ± 0.04 ^ab^	28.7 ± 0.54 ^d^	28.6 ± 0.5 ^cd^	28.1 ± 0.1 ^bc^
Ash	3.6 ± 0.02 ^d^	4.2 ± 0.03 ^e^	4.3 ± 0.05 ^e^	3.1 ± 0.02 ª	3.5 ± 0.04 ^bc^	3.8 ± 0.04 ^d^	3.48 ± 0.02 ^b^	3.92 ± 0.07 ^d^	4.31 ± 0.02 ^e^
CH	35.4 ± 1.4 ^abc^	37.9 ± 1.6 ^bc^	38.8 ± 2.5 ^c^	33.8 ± 1.3 ^a^	34.2 ± 0.8 ^a^	33.9 ± 0.07 ^a^	34.9 ± 0.5 ^ab^	35.1 ± 0.8 ^abc^	36.1 ± 0.8 ^abc^
Ca (mg/100 g)	14.2 ± 1.1 ^a^	86.7 ± 6.0 ^c^	117.8 ± 0.2 ^d^	16.4 ± 0.7 ^a^	74.4 ± 0.4 ^b^	123.0 ± 1.8 ^de^	14.8 ± 0.4 ^a^	78.3 ± 1.2 ^bc^	127.0 ± 3.6 ^e^

The results are expressed as X ± SD, g/100 g on a dry basis. Different superscript letters in the same row indicate significant differences between the samples (*p* ≤ 0.05). N: raw nugget control; N1: raw nugget + 0.75% BF/PH (1:1); N2: raw nugget + 1.25% BF/PH (1:1); FN: fried nugget control; FN1: fried nugget + 0.75% BF/PH (1:1); FN2: fried nugget + 1.25% BF/PH (1:1); BN: baked nugget control; BN1: baked nugget + 0.75% BF/PH (1:1); BN2: baked nugget + 1.25% BF/PH (1:1). CH: carbohydrates; Ca: calcium.

**Table 4 foods-13-01701-t004:** Fatty acid composition of raw, fried, and baked salmon nuggets (without and with the ingredient BF/HP).

	Uncooked Nugget			Fried Nugget			Baked Nugget		
Fatty Acid	N	N1	N2	FN	FN1	FN2	BN	BN1	BN2
C14:0	1.09 ± 0.01 ^cd^	1.17 ± 0.02 ^a^	1.10 ± 0.02 ^bc^	0.91 ± 0.02 ^e^	0.95 ± 0.02 ^e^	0.86 ± 0.00 ^f^	1.10 ± 0.02 ^c^	1.15 ± 0.01 ^ab^	1.04 ± 0.02 ^d^
C16:0	8.84 ± 0.09 ^b^	9.05 ± 0.07 ^a^	8.74 ± 0.09 ^bc^	8.35 ± 0.07 ^d^	8.56 ± 0.07 ^c^	8.19 ± 0.01 ^d^	8.80 ± 0.10 ^b^	8.93 ± 0.00 ^ab^	8.57 ± 0.09 ^c^
C20:0	1.84 ± 0.18 ^c^	1.84 ± 0.06 ^c^	2.16 ± 0.27 ^abc^	2.36 ± 0.07 ^ab^	2.30 ± 0.06 ^ab^	2.54 ± 0.08 ^a^	2.13 ± 0.16 ^bc^	2.22 ± 0.07 ^abc^	2.03 ± 0.08 ^bc^
C22:0	0.81 ± 0.01 ^c^	0.88 ± 0.02 ^a^	0.83 ± 0.02 ^bc^	0.69 ± 0.02 ^de^	0.70 ± 0.01 ^d^	0.64 ± 0.01 ^e^	0.82 ± 0.03 ^bc^	0.85 ± 0.01 ^ab^	0.78 ± 0.02 ^c^
∑SFA	12.60 ± 0.07 ^bd^	12.90 ± 0.06 ^ab^	12.80 ± 0.18 ^bc^	12.30 ± 0.07 ^ef^	12.50 ± 0.06 ^de^	12.20 ± 0.07 ^f^	12.80 ± 0.04 ^b^	13.16 ± 0.08 ^a^	12.40 ± 0.09 ^def^
C16:1	1.49 ± 0.04 ^a^	1.54 ± 0.02 ^a^	1.49 ± 0.02 ^a^	1.22 ± 0.01 ^c^	1.26 ± 0.04 ^c^	1.14 ± 0.01 ^d^	1.50 ± 0.02 ^a^	1.52 ± 0.02 ^a^	1.39 ± 0.03 ^b^
C18:1n-9t	3.52 ± 0.02 ^cd^	3.52 ± 0.01 ^cd^	3.51 ± 0.02 ^d^	3.72 ± 0.01 ^b^	3.77 ± 0.01 ^a^	3.76 ± 0.01 ^a^	3.53 ± 0.02 ^cd^	3.51 ± 0.01 ^d^	3.55 ± 0.01 ^c^
C18:1n-9c	40.90 ± 0.18 ^ab^	41.40 ± 0.11 ^a^	40.70± 0.22 ^bcd^	39.80 ± 0.18 ^ef^	40.20 ± 0.14 ^de^	39.50 ± 0.12 ^f^	40.90 ± 0.06 ^bc^	40.90 ± 0.12 ^bc^	40.50 ± 0.18 ^cd^
C22:1n-9	1.56 ± 0.02 ^abc^	1.63 ± 0.04 ^a^	1.49 ± 0.04 ^c^	1.23 ± 0.04 ^de^	1.29 ± 0.03 ^d^	1.18 ± 0.01 ^e^	1.52 ± 0.06 ^bc^	1.61 ± 0.03 ^ab^	1.50 ± 0.03 ^c^
∑MFA	47.60 ± 0.24 ^ab^	48.01± 0.17 ^a^	47.20 ± 0.29 ^bc^	46.01 ± 0.20 ^ef^	46.60 ± 0.20 ^de^	45.60 ± 0.12 ^f^	47.50 ± 0.12 ^bc^	47.60 ± 0.10 ^ab^	47.01 ± 0.24 ^cd^
C18:2n-6t	2.07 ± 0.25 ^b^	2.33 ± 0.13 ^ab^	2.50 ± 0.31 ^ab^	2.46 ± 0.06 ^ab^	2.50 ± 0.06 ^ab^	2.61 ± 0.08 ^a^	2.49 ± 0.18 ^ab^	2.74 ± 0.09 ^a^	2.39 ± 0.09 ^ab^
C18:2n-6c	28.2 ± 0.09 ^cd^	26.50 ± 0.35 ^e^	27.80 ± 0.47 ^d^	31.18 ± 0.37 ^a^	30.08 ± 0.36 ^b^	31.84 ± 0.01 ^a^	27.64 ± 0.17 ^d^	26.72 ± 0.26 ^e^	29.10 ± 0.44 ^c^
C18:3n-3	3.71 ± 0.05 ^b^	3.89 ± 0.07 ^a^	3.70 ± 0.03 ^b^	2.98 ± 0.07 ^df^	3.09 ± 0.06 ^d^	2.83 ± 0.01 ^f^	3.66 ± 0.07 ^bc^	3.81 ± 0.01 ^ab^	3,.50 ± 0.08 ^c^
C20:2	1.55 ± 0.01 ^cd^	1.69 ± 0.03 ^a^	1.60 ± 0.03 ^bc^	1.33 ± 0.02 ^e^	1.36 ± 0.02 ^e^	1.26 ± 0.01 ^f^	1.58 ± 0.02 ^bc^	1.62 ± 0.02 ^b^	1.50 ± 0.03 ^d^
C20:5n-3	0.74 ± 0.01 ^c^	0.80 ± 0.02 ^a^	0.74 ± 0.02 ^c^	0.61 ± 0.02 ^de^	0.63 ± 0.01 ^d^	0.58 ± 0.01 ^e^	0.75 ± 0.01 ^bc^	0.78 ± 0.01 ^ab^	0.72 ± 0.01 ^c^
C22:6n-3	2.21 ± 0.04 ^b^	2.37 ± 0.10 ^a^	2.25 ± 0.05 ^ab^	1.78 ± 0.05 ^c^	1.81 ± 0.03 ^c^	1.71 ± 0.01 ^c^	2.20 ± 0.08 ^b^	2.20 ± 0.03 ^b^	2.10 ± 0.05 ^b^
∑PUFA	38.49 ± 0.21 ^cd^	37.58 ± 0.25 ^e^	38.62 ± 0.26 ^c^	40.35 ± 0.24 ^a^	39.47 ± 0.25 ^b^	40.80 ± 0.07 ^a^	38.30 ± 0.12 ^cd^	37.70 ± 0.17 ^de^	39.30 ± 0.31 ^b^
∑TransFA	5.59 ± 0.24 ^c^	5.85 ± 0.30 ^bc^	6.01 ± 0.29 ^abc^	6.18 ± 0.06 ^ab^	6.27 ± 0.07 ^ab^	6.37 ± 0.08 ^a^	6.02 ± 0.18 ^abc^	6.26 ± 0.10 ^ab^	5.90 ± 0.09 ^abc^

The results are expressed as X ± SD of the percentage of methyl esters. Different superscript letters in the same row indicate significant differences between the samples (*p* ≤ 0.05). N: raw nugget control; N1: uncooked nugget + 0.75% BF/PH (1:1); N2: raw nugget + 1.25% BF/PH (1:1); FN: fried nugget control; FN1: fried nugget + 0.75% BF/PH (1:1); FN2: fried nugget + 1.25% BF/PH (1:1); BN: baked nugget control; BN1: baked nugget + 0.75% BF/PH (1:1); BN2: baked nugget + 1.25% BF/PH (1:1). ∑SFA: total saturated fatty acids; ∑MFA: total monounsaturated fatty acids; ∑PUFA: total polyunsaturated fatty acids; ∑TransFA: total trans fatty acids.

**Table 5 foods-13-01701-t005:** Cochran Q test results for each attribute and frequency (%) of the CATA test attributes of baked salmon nuggets with different BF/PH contents (0.75, and 1.25%) (*n* = 56).

Attribute	*p*-Value *	BN	BN1	BN2
Bitter taste	0.936	11	13	13
Fatty taste	0.558	16	13	18
Juicy	0.055	16	16	31
Dry	0.422	24	27	18
Faint salmon aroma	0.353	27	36	36
Strong salmon aroma	0.779	13	11	9
Light gold color	0.846	60	60	56
Dark gold color	0.247	4	6	0
Salty	0.338	7	15	9
Low salt	0.417	13	9	7
Crusty	0.892	42	42	38
Faint salmon flavor	0.304	26	15	24
Strong salmon flavor	0.368	33	24	24
Rough interior	0.178	13	22	16
Soft interior	0.565	44	44	51

* *p*-value greater than 0.05 indicates that there are no significant differences at a significance level of 5%.

**Table 6 foods-13-01701-t006:** Cochran Q test results for each attribute and frequency (%) of the CATA test attributes of fried salmon nuggets with different BF/PH contents (0.75, and 1.25%) (*n* = 56).

Attribute	*p*-Value *	FN	FN1	FN2
Bitter taste	0.646	14 ^a^	18 ^a^	12 ^a^
Fatty taste	0.307	20 ^a^	22 ^a^	28 ^a^
Juicy	0.018	6 ^a^	22 ^ab^	26 ^b^
Dry	0.007	33 ^b^	24 ^ab^	10 ^a^
Faint salmon aroma	0.756	35 ^a^	39 ^a^	41 ^a^
Strong salmon aroma	0.060	8 ^a^	18 ^a^	6 ^a^
Light gold color	0.458	35 ^a^	45 ^a^	45 ^a^
Dark gold color	0.878	24 ^a^	22 ^a^	20 ^a^
Salty	0.641	16 ^a^	12 ^a^	12 ^a^
Low salt	0.097	12 ^a^	10 ^a^	20 ^a^
Crusty	0.141	49 ^a^	65 ^a^	59 ^a^
Faint salmon flavor	0.482	20 ^a^	29 ^a^	26 ^a^
Strong salmon flavor	0.193	26 ^a^	22 ^a^	14 ^a^
Rough interior	0.405	29 ^a^	20 ^a^	26 ^a^
Soft interior	0.237	29 ^a^	41 ^a^	41 ^a^

* *p*-value in bold indicates that there are significant differences at a significance level of 5%. Different superscripts (lowercase letters ^a^, ^b^) indicate significant differences between samples at a significance level of 5%.

## Data Availability

The data presented in this study are available on request from the corresponding author. The data are not publicly available due to the unavailability of a public repository at the Department of Nutrition, Faculty of Medicine, University of Chile.

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
