# Peer review of "Valorization of the Salmon Frame as a High-Calcium Ingredient in the Formulation of Nuggets: Evaluation of the Nutritional and Sensory Properties"

_foods, 2024, doi:10.3390/foods13111701_

Round 1

Reviewer 1 Report

Comments and Suggestions for Authors

The salmon is an ideal bio-resource for food industry. The study of the salmon frame nutrition is beneficial in maximising its utilisation. The manuscript discloses the salmon frame as a high calcium ingredient in the formulation of nuggets in the view of the nutritional and sensory properties. The purpose of the study is well articulated in the introduction. The well-written manuscript focuses on an interesting and topical subject and is of considerable practical importance; however, the manuscript requires a few additional revisions.

(1)Could authors provide more information/references on how salmon by-products be handled, recycled and utilised in Chile?

(2)Have you analyzed the chemical properties of sunflower oil? What impact does the use of sunflower oil have on the nutritional composition and fatty acid profile of the finished product?

(3)Would adding higher contents of Bone Flour (BF) and Protein Hydrolysate (PH) affect the quality of the finished product?

(4)Have you analyzed the physical properties of the salmon nuggets?

(5) Is frying or baking the best way to process? Does it maximise nutritional value?

(6)It is hoped that the authors will provide further discussion with reference to the recently revised NO. 211 of the Reglamento Sanitario de los Alimentos (Food Hygiene Regulation), which states that Chilean companies are required to add vitamin D to the milk and bread that they produce. The question of how to optimise calcium supplementation in the development of products in the future is not limited to the use of calcium by taking nutritional supplements, medications, or by changing other means such as lifestyle habits.

Author Response

Response to Reviewer 1 Comments

Manuscript ID: foods-1042209

We thank this reviewer for his/her positive evaluation.

Point 1: Could authors provide more information/references on how salmon by-products be handled, recycled and utilised in Chile?

Response: More information on the destination of by-products from the fishing industry, including the salmon industry, has been considered. Line 46 to 48 the following text has been added “The main destination of the byproducts of the fishing industry is the production of fishmeal and fish oil. It is estimated that between 27 and 48% of the total production of fishmeal and fish oil are obtained from byproducts (FAO, 2022)”.

Point 2: Have you analyzed the chemical properties of sunflower oil? What impact does the use of sunflower oil have on the nutritional composition and fatty acid profile of the finished product?

Response: Yes. The fatty acid composition of the oil was analyzed. The major ones are cited in the text (L406-410): pre-frying of nuggets carried out with sunflower oil (Natura®), which mainly contains linoleic and oleic acid (52.0 and 37.7%, respectively). Therefore, the high contents of oleic and linoleic acids in all types of salmon nuggets in our study could be attributed to the fatty acid composition of the salmon fillets and the amount of oil absorption during pre-frying of nuggets [36, 39, 40] carried out with sunflower oil (Natura®).

Point 3: Would adding higher contents of Bone Flour (BF) and Protein Hydrolysate (PH) affect the quality of the finished product?

Response: Yes. According to the analysis of calcium content (L334-336), it can be established that if the percentage of the incorporated ingredient is increased, the calcium content in the nuggets will increase, but not the protein content (L298-299). Additionally, the sensory properties of the product may decrease (L512-515)

Point 4: Have you analyzed the physical properties of the salmon nuggets?.

Response: In this study, the physical properties of the nuggets were not analyzed.

Point 5: Is frying or baking the best way to process? Does it maximise nutritional value?

Response: The following paragraph was added in the text (L325-328): According to the results of the proximal chemical analyses of the nuggets, it can be concluded that, in general, the incorporation of BF:PH (1:1) did not significantly affect their macronutrient content. In contrast, the cooking method, such as shallow frying, increased the lipid content compared to the baked nuggets.

Point 6: It is hoped that the authors will provide further discussion with reference to the recently revised NO. 211 of the Reglamento Sanitario de los Alimentos (Food Hygiene Regulation), which states that Chilean companies are required to add vitamin D to the milk and bread that they produce. The question of how to optimise calcium supplementation in the development of products in the future is not limited to the use of calcium by taking nutritional supplements, medications, or by changing other means such as lifestyle habits.

Response: The following paragraph was added in the text (L355-363): According to these results, it can be established that the incorporation of salmon bones allowed the calcium content of the nuggets to increase. Thus, the revaluation of by-products from the food industry and their incorporation into foods could be considered a potential strategy to increase calcium intake in the Chilean population, which is below the established requirements even when there are supplements and frequent recommendations for a healthy lifestyle. A concrete example of the incorporation of micronutrients in foods in our country is given by the recent modification in legislation, which establishes that dairy products and flour must be supplemented with vitamin D (art 211, 216, 350) (52) due to a deficiency cross-sectional vitamin D in the Chilean population.

Reviewer 2 Report

Comments and Suggestions for Authors

I reviewed the manuscript “Valorization of the salmon frame as a high calcium ingredient in the formulation of Nuggets: Evaluation of the nutritional and sensory properties” (foods-3024415). The authors proposed an interesting idea to take advantage of residues from salmon industry: Bone flour and Protein hydrolysate were previously reported by the same group, and the manuscript is an extension of the previous work.

General comments:

Tittle includes calcium modification; however, it is included in Introduction and Results information regarding protein content. It seems if it was also an objective not achieved, in fact it is strange the protein content was lower if protein hydrolysate were added. Also, it is important to include if protein diminution is acceptable for nuggets. I suggest modifying introduction and results section to be congruent between Tittle and manuscript content.

Please check decimal punctuation in all the manuscript, it is used indiscriminately “point” and “commas”. It is also important to check significant figures in all the manuscript.

Specific comments:

Abstract, Line 21. The phrase “The objective was to 21 evaluate the effect of the incorporation of BF and PH in a 1:1 ratio…” is not clear. It important to include the two concentrations added in nuggets formulation.

Section 2.2.1. Formulation is incomplete, it must be included the full formulation in order to understand the following discussions. Maybe a Table would be useful to understand the changes in each formulation.

Line 153. Please check BF3 nomenclature.

Section 2.2.4. It must be included a short description of the technique. Acid digestion (open, close?)

Section 3.5.2. must be edited, its similar the text in both cases (BNs and FN samples). It could be reduced and include the Data Tables in a Supplementary file.

Author Response

Response to Reviewer 2 Comments

We thank this reviewer for his/her positive evaluation.

Point 1: The title includes calcium modification; however, it is included in the introduction and results information regarding protein content. It seems if it was also an objective not achieved, in fact it is strange the protein content was lower if protein hydrolysate were added. Also, it is important to include if protein diminution is acceptable for nuggets. I suggest modifying introduction and results section to be congruent between Tittle and manuscript content.

Response: The objective of this research was to increase the calcium content of the nuggets, but not the protein content. The protein hydrolysate was added to the formulation to improve the absorption of calcium in the body, according to what is described in the literature. This objective was addressed in another work that is simultaneously under review in the same journal (Manuscript ID: foods-3012689: Fish bones as calcium source: bioavailability of micro and nanoparticles). Considering his observation, the objective of this work and the summary were modified. Also added some comments in the results and discussion section. The protein content in this study is similar to the commercial salmon nuggets in Chile.

Abstract, Line 21. The phrase “The objective was to evaluate the effect of the incorporation of BF and PH in a 1:1 ratio…” is not clear. It important to include the two concentrations added in nuggets formulation.

Response: The objective was modified in the text (L94-97): The aim was to evaluate the effect of the incorporation of BF and PH in a 1:1 ratio (providing two calcium concentrations to the nuggets, 75 and 125 mg/100g), obtained from the salmon frame, on the calcium content and sensory attributes of salmon nuggets submitted to baking or shallow frying

Section 2.2.1. Formulation is incomplete, it must be included the full formulation in order to understand the following discussions. Maybe a Table would be useful to understand the changes in each formulation.

Response: A table with ingredients for each formulation was added (L123-141)

Line 153. Please check BF3 nomenclature.

Response: The name of the reagent was added to the text (L182): Boron Trifluoride (BF3)

Section 2.2.4. It must be included a short description of the technique. Acid digestion (open, close?)

Response: The following paragraph was added in the text (L191-198): About 2 grams of previously dried nuggets were calcined in a muffle furnace at 500ºC for 8 h. The ashes were then digested with 2 ml of concentrated nitric acid, allowed to evaporate on a hot plate, and placed in the muffle furnace at 500ºC for 1 hour. The ashes were then dissolved in 10 mL of 1 N HCl, heated on a hot plate, and transferred to a 10 mL volumetric flask. Additional dilutions were performed with deionized water to bring the concentrations into the linear range of atomic absorption spectroscopy. Lanthanum nitrate was added to the final dilution to achieve a concentration of 0.1% lanthanum.

Section 3.5.2. must be edited, its similar the text in both cases (BNs and FN samples). It could be reduced and include the Data Tables in a Supplementary file.

Response: Tables were eliminated from the main text, and the results were more explained in the text.

Round 2

Reviewer 2 Report

Comments and Suggestions for Authors

The authors considered almost all the suggestions.